# Exploring the Role of Endoplasmic Reticulum Stress in Hepatocellular Carcinoma through mining of the Human Protein Atlas

**DOI:** 10.3390/biology10070640

**Published:** 2021-07-09

**Authors:** Nataša Pavlović, Femke Heindryckx

**Affiliations:** Department of Medical Cell Biology, Uppsala University, 751 23 Uppsala, Sweden; natasa.pavlovic@mcb.uu.se

**Keywords:** endoplasmic reticulum stress, unfolded protein response, hepatocellular carcinoma

## Abstract

**Simple Summary:**

Hepatocellular carcinoma is a highly deadly primary liver cancer. It is usually diagnosed at a late stage, when therapeutic options are scarce, and the lack of predictive biomarkers poses a challenge for early detection. A known hallmark of hepatocellular carcinoma is the accumulation of misfolded proteins in the endoplasmic reticulum (ER), known as ER-stress. Growing experimental evidence suggests that ER-stress is involved in liver cancer initiation and progression. However, it remains unclear if ER-stress markers can be used as therapeutic targets or biomarkers for patients with liver cancer. In this study, we evaluated the prognostic value of proteins involved in managing ER-stress in liver cancer by mining a publicly available patient-derived database, the Human Protein Atlas. We thereby identified 44 ER-stress-associated proteins as prognostic markers in liver cancer. Furthermore, we discussed the expression of these markers in relation to disease stage, age, sex, ethnicity, and tissue localization.

**Abstract:**

Endoplasmic reticulum (ER) stress and actors of unfolded protein response (UPR) have emerged as key hallmarks of hepatocarcinogenesis. Numerous reports have shown that the main actors in the UPR pathways are upregulated in HCC and contribute to the different facets of tumor initiation and disease progression. Furthermore, ER-stress inducers and inhibitors have shown success in preclinical HCC models. Despite the mounting evidence of the UPR’s involvement in HCC pathogenesis, it remains unclear how ER-stress components can be used safely and effectively as therapeutic targets or predictive biomarkers for HCC patients. In an effort to add a clinical context to these findings and explore the translational potential of ER-stress in HCC, we performed a systematic overview of UPR-associated proteins as predictive biomarkers in HCC by mining the Human Protein Atlas database. Aside from evaluating the prognostic value of these markers in HCC, we discussed their expression in relation to patient age, sex, ethnicity, disease stage, and tissue localization. We thereby identified 44 UPR-associated proteins as unfavorable prognostic markers in HCC. The expression of these markers was found to be higher in tumors compared to the stroma of the hepatic HCC patient tissues.

## 1. Introduction

Hepatocellular carcinoma (HCC) is a the most common type of primary liver cancer and usually develops in a background of chronic liver disease and inflammation. With global HCC incidence and mortality consistently rising, advanced screening programs, novel predictive biomarkers, and new therapeutic options for advanced HCC are desperately needed [1]. Further research on the molecular pathogenesis of HCC progression is therefore encouraged, as this could eventually lead to the discovery of new biomarkers for improved HCC surveillance and prevention, as well as new therapeutic targets for treating patients with liver cancer.

The unfolded protein response (UPR) is a conserved cellular survival strategy which is activated in response to the accumulation of unfolded or misfolded proteins in the endoplasmic reticulum (ER) lumen, an event known as ER-stress. Factors such as pathogens, mutations, nutrient deprivation, hypoxia, and an increased metabolic rate lead to an increase in the cell’s protein secretory load. This impairs the ER’s capacity to accurately monitor protein folding and synthesis, which induces UPR signaling through three major ER transducers—IRE1α, PERK, and ATF6—with the goal of reestablishing ER homeostasis. However, sustained ER-stress and prolonged UPR signaling can activate pro-apoptotic signaling pathways [2,3].

Chronic ER-stress and increased UPR activity have long been reported as the hallmarks of solid tumors, including HCC. Upregulated UPR signaling has been noted in the majority of human HCCs, irrespective of grade or stage [4]. Our group recently reported that inhibiting the IRE1α-mediated ER-stress pathway significantly reduced tumor burden in a chemically induced fibrotic HCC mouse model [5]. This pathway was also found to be crucial for liver cirrhosis progression in vivo [6]. The PERK- and ATF6-mediated pathways have also been widely implicated in HCC progression by increasing HCC chemoresistance and promoting tumor invasiveness and adaptation to hostile microenvironmental conditions [7,8]. However, despite mounting reports on the involvement of ER-stress in the different facets of HCC pathogenesis and the success of ER-stress inhibitors and inducers in pre-clinical HCC models [9], the translational potential of targeting ER-stress actors has yet to be fully assessed and utilized. Therefore, in the present study, our aim was to perform a systematic overview of UPR-associated proteins as prognostic markers in HCC through mining the publicly available Human Protein Atlas database [10,11]. Here, we showed that the expression of 44 UPR-associated proteins significantly correlated with poor prognosis in patients with HCC, marking them as clinically relevant biomarkers in the treatment of liver cancer. Furthermore, we presented and discussed the expression of these markers in relation to disease stage, age, sex, ethnicity, and tissue localization in an attempt to better understand how the detrimental effect ER-stress imposes on hepatocarcinogenesis could be effectively targeted and utilized in HCC treatment.

## 2. Materials and Methods

### 2.1. Selection of UPR Genes

A gene-set of 129 proteins involved in the UPR was generated from online databases: the Harmonizome [12,13] and STRING [14,15] (Figure 1 and Figure 2, Appendix A). The publicly available database from the Human Protein Atlas was then used to determine which of these genes were prognostic markers in patients with liver cancer [10,11] (Figure 3).

### 2.2. The Human Protein Atlas Data Mining

Patient survival was correlated to the expression levels of the selected 129 UPR-associated markers by using publicly available data from the Human Protein Atlas. Patients were classified into two expression groups based on the fragments per kilobase of exon per million reads (FPKM) value of each gene. The expressions of UPR-associated markers in the different tumor stages of HCC were taken from the Human Protein Atlas database using the RNA sequence data from 365 patients derived from the Cancer Genome Project [16].

### 2.3. Evaluating the Tumor/Stroma Expression of Unfavorable HCC Prognostic Markers

Images of healthy livers and HCC biopsies stained with antibodies against the UPR-associated unfavorable HCC prognostic markers were derived from the Human Protein Atlas. Five replicates stained with the antibody with the strongest staining intensity were selected for each marker. The staining intensity was then scored on a scale from 0 to 3, with 0 corresponding to no staining detected, 1 to weak staining, 2 to moderate staining, and 3 to the strongest staining intensity on healthy, stromal, and tumoral tissues. The biopsy staining was scored by the authors independently, after which the average scores were generated.

### 2.4. Statistical Analysis

The dataset derived from the Human Protein Atlas was imported into GraphPad Prism 8. The statistical significance between different experimental groups (age, sex, racial ethnicity, and HCC stage) was determined through a two-way ANOVA using the Dunn–Šidák correction for multiple comparisons. A volcano plot was generated after performing a multiple t-test on the different stages of HCC in order to visualize the ER-stress genes with the highest fold-change and statistically significant differences. The correlation between the biomarker expression level and patient survival was examined using GraphPad Prism 8. Based on the data from the Human Protein Atlas, the FPKM value of the appropriate gene was used as a cut-off to determine high or low expression. The FPKM/cut-off value corresponded to the best expression cut-off level, meaning that the FPKM value that would yield the maximum difference in regard to survival between the two groups at the lowest log-rank *p*-value. Statistical significance between the groups of the Kaplan–Meier curves was determined with a log-rank test. *p*-values of <0.05 were considered statistically significant.

## 3. Results

### 3.1. The Predictive Value UPR-Associated Protein Expression in HCC

A systematic overview of UPR-associated proteins as prognostic markers for HCC was performed using the publicly available Human Protein Atlas database (Figure 1). This search identified 44 out of 129 selected ER-stress genes as unfavorable prognostic markers in HCC and 1 as a favorable prognostic marker (Figure 2, Figure 3 and Figure 4, Appendix A). We compared the mRNA expression of these 45 markers in the different stages of HCC, which revealed that the majority of UPR-associated genes had the highest expression in disease stages 2 and 3 (Figure 4A,B). Notably, the expression of markers RACK1, ATF4, GANAB, PDIA6, SSR2, EIF2S3, CHOP, TRIB3, and CALU, known to be strongly associated with HCC invasiveness and chemoresistance [17,18,19,20], was shown to significantly increase with the progression of HCC to stages 2 and 3 (Figure 4D–L). The correlation between the expression of these biomarkers and patient survival is shown in Figure 5. It should, however, be pointed out that the decreased expression of prognostic markers in stage 4 could be explained by there being less available patient-derived data from this HCC stage.

### 3.2. The Expression of UPR-Associated Markers in Relation to Sex, Age, and Ethnicity in HCC

Cancer progression and treatment efficacy are known to be impacted by different host characteristics, including patient sex and age [21]. Furthermore, the etiological backgrounds of HCC vary significantly between the Western world and East Asia [22]. Therefore, through mining the Human Protein Atlas database, we aimed to evaluate the potential correlation between the expression of the UPR-associated unfavorable prognostic markers in HCC with patient sex, age, and ethnicity. Overall, no significant differences were noted when comparing the expression of UPR markers between sexes in HCC patients, with the exception of the markers EIF2S3, SSR2, and ATF4, where a higher expression was observed in female patients (Figure 6A,B). Similarly, no notable differences were observed when comparing the UPR marker expressions between three different age groups (<40, 41 to 65, and >66; Figure 7A,B), with the exception of the markers RACK1, ATF4, and CHOP (Figure 7C,D,K). Finally, when comparing the mRNA expression of the UPR markers in HCC between three ethnic categories (Asian, Black or African American, and White; Figure 8), differences in expression were observed for markers RACK1, ATF4, and SSR2, with the common trend of the expression being elevated in the Asian group compared to the Black or African American and White groups (Figure 8C,D,G).

### 3.3. Tumor/Stroma Expression of UPR-Associated Unfavorable Prognostic Markers in HCC

Metabolic and oncogenic abnormalities in both tumor cells and the surrounding stromal microenvironment are known inducers of ER-stress [23]. HCC is characterized by abundant stroma and develops in a background of sustained inflammation and fibrosis. Therefore, tumor–stroma interactions are a key regulator of HCC pathogenesis and response to treatment [1]. In order to determine the tumor–stroma distribution of the UPR-associated biomarker expression in the patient HCC tissue, we scored the immunohistochemical staining intensity on HPA-derived images of healthy, stromal, and tumoral liver tissue (Figure 9) [24,25,26,27,28,29,30,31,32,33,34,35,36,37,38,39]. Notably, we observed a trend of increased UPR marker protein expression in the tumorous tissue compared to stromal compartments, where the expression was overall downregulated to the baseline levels of the healthy tissue (Figure 9A). Furthermore, upregulated protein expression in the tumorous tissue was observed for the actors of the eIF2α-ATF4- CHOP pathway, as well as markers such as RACK1, PDIA6, and ERO1α, all of which have been proposed as potential therapeutic targets in HCC [17,20,40,41].

## 4. Discussion

The actors of the UPR have been widely demonstrated to influence different facets of HCC progression. While numerous preclinical models have proposed ER-stress inhibitors and inducers as a promising therapeutic strategy for patients with liver cancer, the translational potential of targeting UPR signaling is still unclear. In this study, we investigated the prognostic value of UPR-associated proteins in liver cancer through mining the Human Protein Atlas database. Furthermore, in an effort to potentially identify a subpopulation of liver cancer patients that could benefit from ER-stress-targeted therapy, the expression of these markers was evaluated in relation to patient sex, age, ethnicity, and disease stage. The UPR-associated protein expression was also discussed in regard to its hepatic tissue localization.

Through mining the Human Protein Atlas, 44 UPR-associated proteins were identified as unfavorable prognostic markers in HCC. While the majority of the UPR markers that were found to be prognostic could not be categorized under one specific UPR branch, several observations could be made regarding their upregulation, correlating with poor HCC prognosis. For instance, unfavorable prognostic markers belonging to the PERK-mediated UPR pathway, such as eIF2α, ATF4, GADD34, and CHOP, have been widely reported as key regulators of HCC chemosensitivity and apoptosis in targeted HCC therapy. Notably, ATF4, CHOP, and eIF2 subunit gamma (EIF2S3) expression were found to significantly increase with the advancement of HCC to later stages, in line with the possible role of these markers in HCC chemoresistance [18,42,43,44]. Moreover, CHOP was shown to be strongly implicated in hepatic inflammation, fibrosis, and hepatocarcinogenesis, while CHOP-null mice were found to be resistant to chemically induced HCC [45]. Another unfavorable marker mediated by the PERK ER-stress branch, ER resident oxidase 1-alpha (ERO1-α), has been found to play a key role in HCC invasion, metastasis, and angiogenesis by triggering the S1PR1/STAT3/VEGF-A signaling pathway, both in vitro and in vivo [41].

Receptor of activated protein kinase 1 (RACK1) was shown to play a crucial role in the activation of IRE1α signaling in response to the sorafenib treatment of HCC cells, as RACK1 overexpression led to the increased phosphorylation of IRE1α and enhanced XBP1 splicing, thereby protecting HCC cells from sorafenib-induced apoptosis [17]. Moreover, our analysis of RACK1 expression between the different stages of HCC showed a significant increase between stage 1 and stages 2 and 3, adding to the relevance of potentially targeting RACK1 in advanced HCC.

Signal sequence receptor subunit 2 (SSR2) was found to be overexpressed in patient HCC tissue, with increased expression significantly correlating with disease progression. Hong et al. recently reported that SSR2was involved in the proliferation and invasion of HCC cells, as SSR2 knockdown was shown to suppress the epithelial mesenchymal transition (EMT) of HCC cells [19].

Several homologs of the Hsp40/DnaJ protein family (DnaJB6, DnaJC1, DnaJC5, DnaJC7, and DnaJC21) have been identified as unfavorable prognostic markers in liver cancer. Wang et al. recently demonstrated a novel oncogenic function of DnaJC5 in promoting HCC cell proliferation by regulating the SKP2-mediated degradation of tumor suppressor p27 [46]. Furthermore, other members of the DnaJ protein family have been assigned roles in other liver pathologies, such as steatosis development for DnajC7 [47] and liver cirrhosis for DnajC21 [48]. Several members of the DnaJ protein family have been determined to act as co-chaperones of Hsp70 proteins through different mechanisms, such as binding immunoglobulin protein (BiP). For instance, DnaJC1 has been shown to regulate translation in a BiP-dependent manner by binding its cytosolic domain to the 80S ribosome, thereby ensuring BiP interaction with nascent polypeptide chains entering the ER [49]. The upregulation of DnaJ proteins in HCC therefore aligns with a number of studies showing an increased expression of BiP in HCC patients [7,50].

Notably, our analysis identified one UPR-associated protein, phosphoinositide-3-kinase regulatory subunit 1 (PIK3r1), as a favorable prognostic marker in liver cancer. Winnay et al. reviously reported on an ER-stress-dependent interaction between PIK3r1, a key regulator of insulin action, and nuclear XBP1 accumulation, followed by UPR gene induction. This study demonstrated that PIK3r1 deletion in the liver attenuates adaptive UPR signaling and promotes inflammation [51]. We could therefore speculate that increased PIK3r1 expression is involved in a pro-adaptive response to HCC-induced ER-stress, whereby it suppresses inflammation and exerts an anti-carcinogenic function. However, given its function in metabolomic homeostasis, the role of PIK3r1 in HCC-associated ER-stress could be largely dependent on disease etiology, and this warrants further investigation.

Overall, the UPR-associated proteins that were identified as unfavorable prognostic markers in HCC in our study are in line with previous reports that confirm their contribution to ER-stress-mediated hepatocarcinogenesis and their potential as therapeutic targets in HCC. These findings therefore strengthen the evidence of their involvement in HCC progression and their translational potential for patients with liver cancer.

When comparing the expression of the UPR-associated unfavorable prognostic markers in relation to patient sex, no notable differences were observed. This finding is in contrast to reports on the sexual dimorphism of ER-stress susceptibility in different in vivo models. Male prepubertal rats were shown to have a lower predisposition to hepatic ER-stress, while an increase in UPR marker expression in males was noted during the transition from a prepubertal to an adult age, which was strongly linked to the notable increase in testosterone levels in this phase [52]. Similarly, Hodeify et al. reported on sex differences in ER-stress-induced acute kidney injury and observed a significantly higher susceptibility in male mice compared to females [53].

Similarly, we did not observe significant differences when we compared the UPR marker expressions between the three different age groups, with the exception of the markers RACK1, ATF4, and CHOP. While an age-related decline in RACK1 expression has been noted in several studies on age-associated immunosenescence [54,55], the increased expression of ATF4 in the liver was found to correlate with a longer life span in different strains of slow-aging mice [56]. This finding aligns with the diverse pro-adaptive ATF4-mediated UPR pathways that could play a role in controlling the aging rate and UPR maintenance. Contrary to our findings, an age-related upregulation of CHOP has been reported by previous studies, where it was also noted that increased CHOP expression sensitized aging hepatocytes to oxidative injury, ER-stress, and autophagy [57,58,59]. Overall, our observation of a lack of age-related increase in ER-stress marker expression in HCC patients was contrary to reports of age-related deterioration of the UPR machinery and an increase in protein misfolding, which is known to exacerbate existing diseases, including different types of cancer [60].

When we compared the mRNA expression of the UPR markers in HCC between three ethnic categories (Asian, Black or African American, and White), differences in expression were observed for the markers RACK1, ATF4, and SSR2. While a common feature of HCC is that it arises in the background of chronic liver disease and cirrhosis, dominant risk factors for HCC development differ between different ethnicities. In East Asia (with the exception of Japan) and Southeast Asia, these risk factors are the hepatitis B virus HBV and aflatoxin exposure, while in Europe and North America they include the hepatitis C virus and the metabolic syndrome [22]. These distinct HCC etiologies could in part explain the differences in UPR-marker expression we observed between different ethnic groups, but also highlight the importance of etiologic factors when evaluating biomarkers and therapeutic targets in HCC.

Finally, a notable trend of increased UPR marker protein expression in the tumorous hepatic tissue was observed compared to stromal compartments, where the expression was overall downregulated to the baseline levels of the healthy tissue. For instance, key ER-stress markers associated with HCC chemoresistance, such as ATF4, ERO1-α, EIF2S3, and CHOP, were found to have higher expression in tumoral tissue compared to the stromal compartment of the liver biopsies. These ER-stress regulators have also been proposed as therapeutic targets for inducing HCC apoptosis and sensitizing HCC cells to chemotherapy. Chronic UPR activation and ER-stress occurrence in tumor cells is known to be a result of an increased metabolic rate and a mutation-driven protein synthesis need [61]. However, ER-stress in stromal cells has also been reported as a key contributor to hepatocarcinogenesis. In our previous paper, we showed that ER-stress markers belonging to the IRE1α branch were mainly present in the hepatic stellate cell population in the livers of mice with HCC. This increased expression was shown to correlate with stellate cell activation, pro-fibrotic signaling, and ultimately tumor progression in HCC. Furthermore, we reported that the expression of IRE1α signaling pathway components was upregulated in fibrotic HCC patient tissue compared to non-fibrous HCC tissue, while this increased expression significantly correlated with poor survival in patients with liver cancer [5]. Moreover, we recently reported that ER-stress occurrence in macrophages was correlated with an anti-tumoral macrophage phenotype and the increased macrophage clearance of HCC cells [62]. These findings emphasized the contribution of ER-stress and UPR activation in stromal tissue to HCC progression, as well as the therapeutic potential of targeting stromal cells undergoing ER-stress in liver cancer.

This insight on the tumor/stroma distribution of UPR marker expression in HCC could prove relevant when assessing the use and development of nanotherapies targeted at these markers. Delivering the ER-stress inducers or inhibitors specifically to the cell-of-interest could be of great importance, as systemically targeting UPR pathways could have long-term undesired side-effects for patients.

## 5. Conclusions

Broad experimental evidence from preclinical and patient studies highlights ER-stress as an important contributor to the hepatocarcinogenic process. While it is important to shed light on the molecular implications of ER-stress markers for the progression of HCC, there is a great need to explore the translational potential and the clinical relevance of these findings. By investigating the predictive value of ER-stress markers in HCC, we identified 44 proteins as unfavorable for patients with liver cancer. Through quantifying the distribution of the expression of these markers in patient HCC tissue as well as discussing the current literature on how these markers are involved in HCC pathogenesis, we provide insight into how the therapeutic value of targeting ER-stress in HCC can be utilized for the advancement of liver cancer prevention and treatment.

## Figures and Tables

**Figure 1 biology-10-00640-f001:**
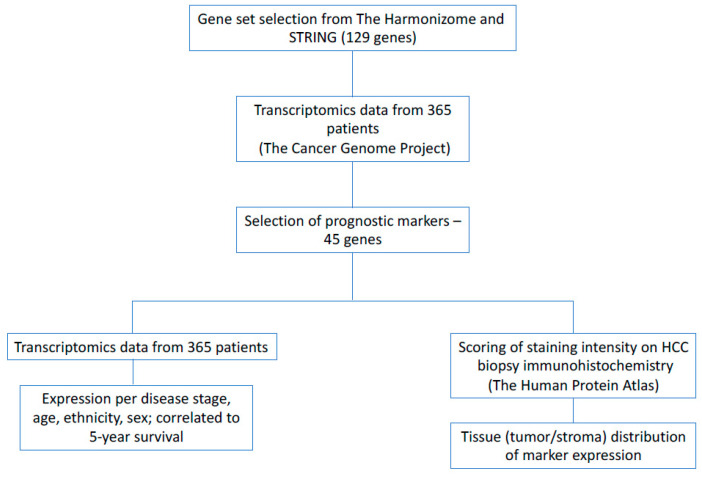
The method used for selecting UPR-associated genes, determining their prognostic value in HCC, and comparing and presenting their expression to different patient variables.

**Figure 2 biology-10-00640-f002:**
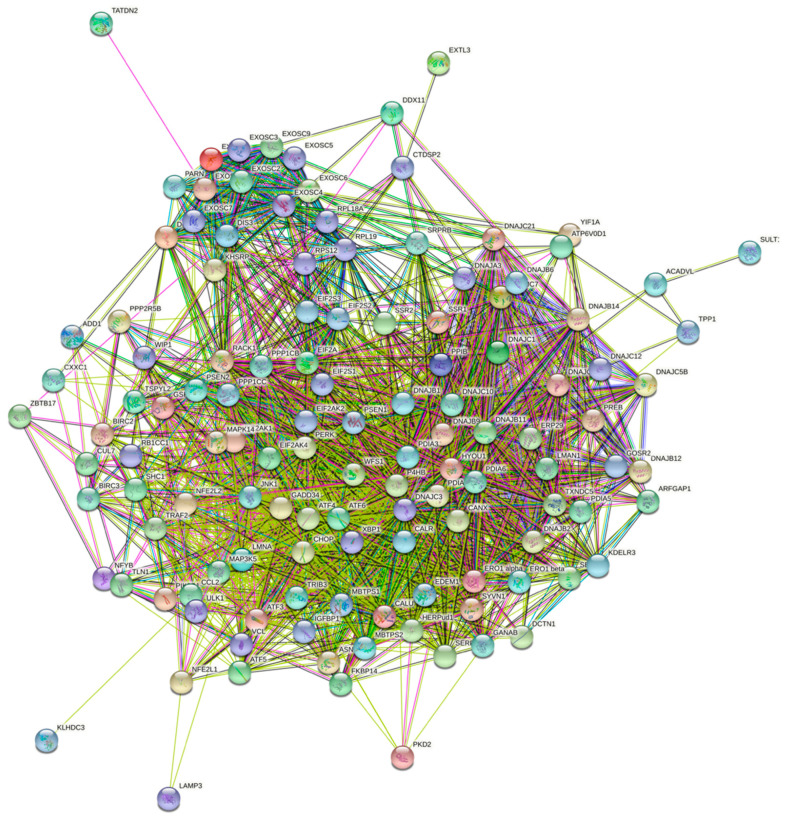
An interconnected gene map of the entire UPR-associated gene-set derived from STRING and the Harmonizome.

**Figure 3 biology-10-00640-f003:**
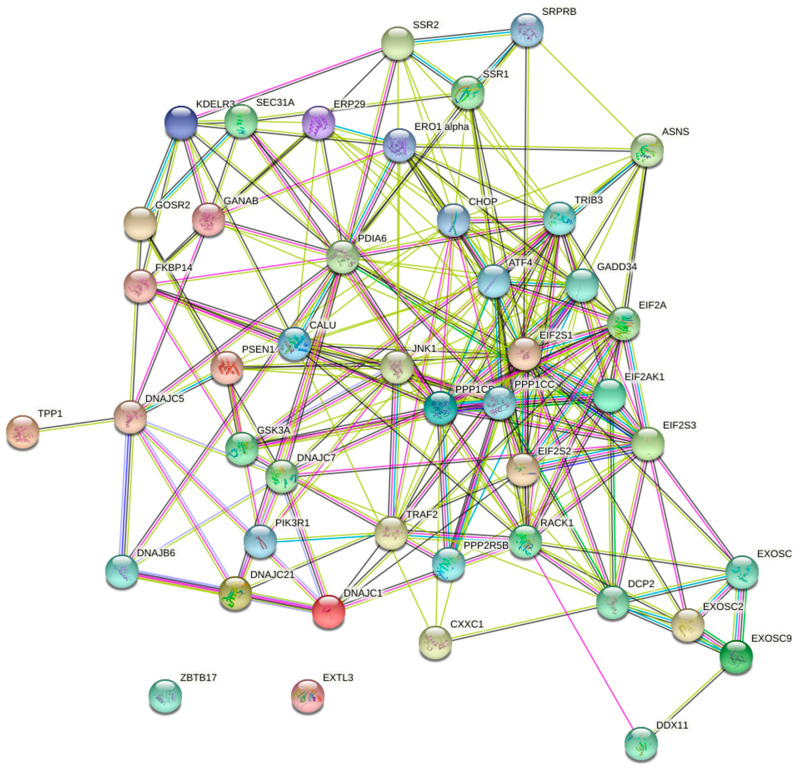
An interconnected gene map of UPR-associated genes with prognostic value in HCC.

**Figure 4 biology-10-00640-f004:**
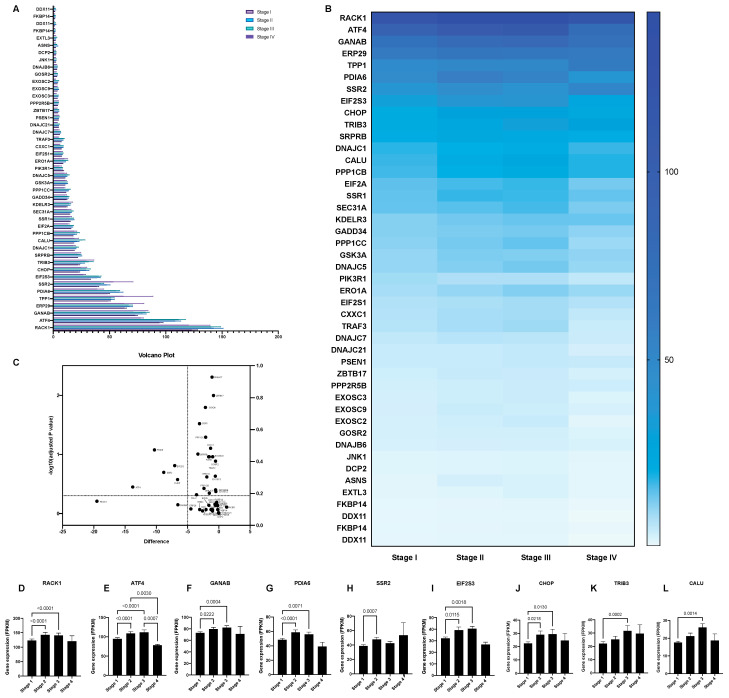
The mRNA expression of unfavorable UPR-associated markers per HCC stage shown in a bar chart (**A**), a heatmap (**B**), and a volcano plot (**C**). Markers with statistically significant expression levels between the different stages are shown as individual bar charts (**D**–**L**). *p*-values of <0.05 were considered statistically significant.

**Figure 5 biology-10-00640-f005:**
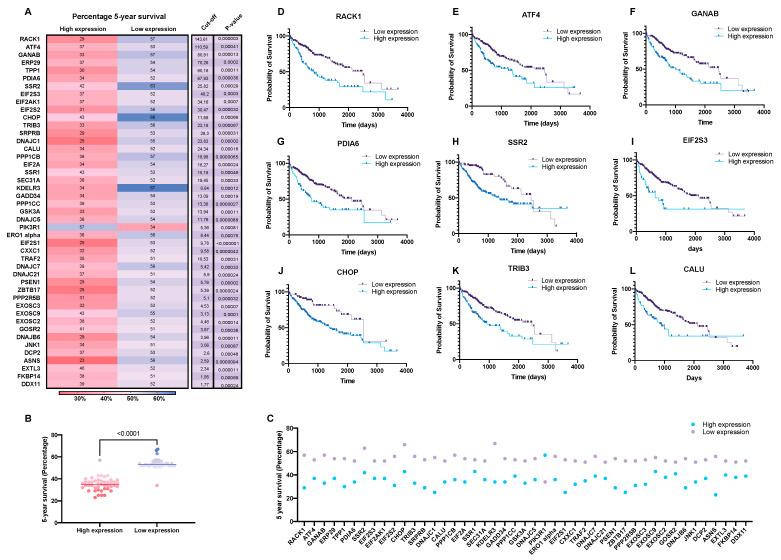
Five-year HCC patient survival data correlated to the expression of unfavorable UPR-associated markers (**A**–**C**). The selected markers are shown as individual Kaplan–Meier curves (**D**–**L**).

**Figure 6 biology-10-00640-f006:**
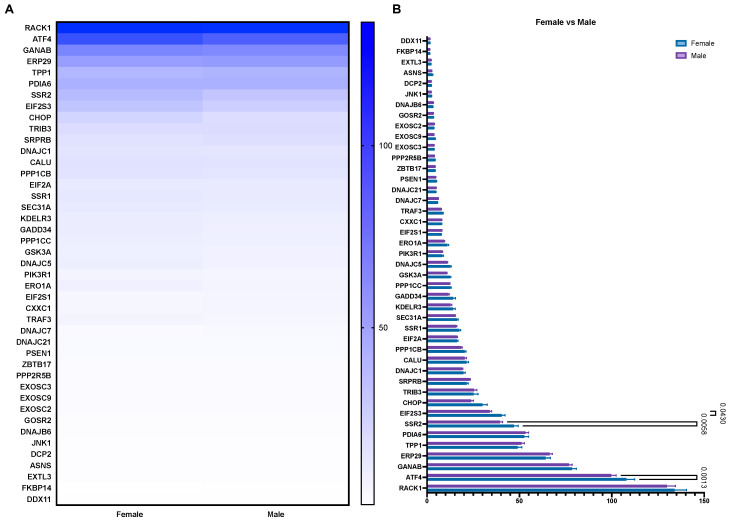
The mRNA expression of unfavorable UPR-associated markers per sex shown in a heatmap (**A**) and a bar chart (**B**).

**Figure 7 biology-10-00640-f007:**
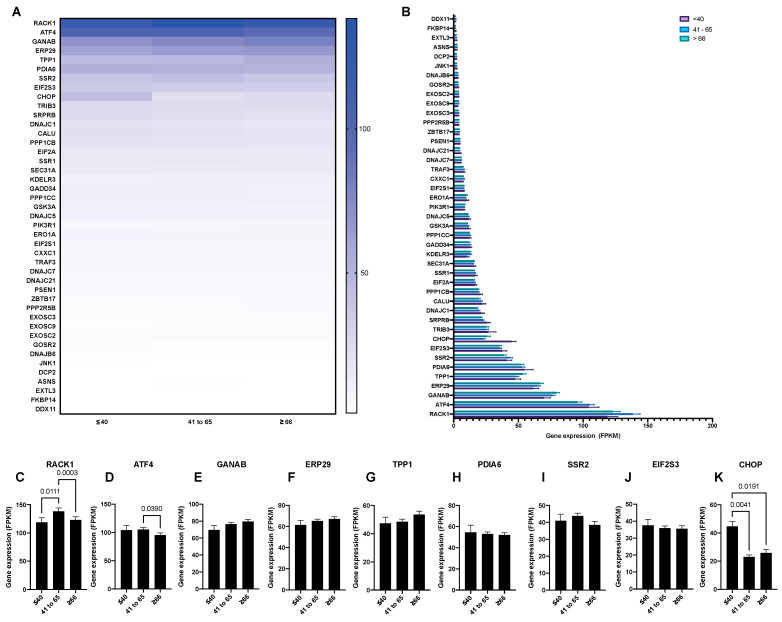
The mRNA expression of unfavorable UPR-associated markers per age group shown in a heatmap (**A**) and a bar chart (**B**). The selected markers are shown as individual bar charts (**C**–**K**).

**Figure 8 biology-10-00640-f008:**
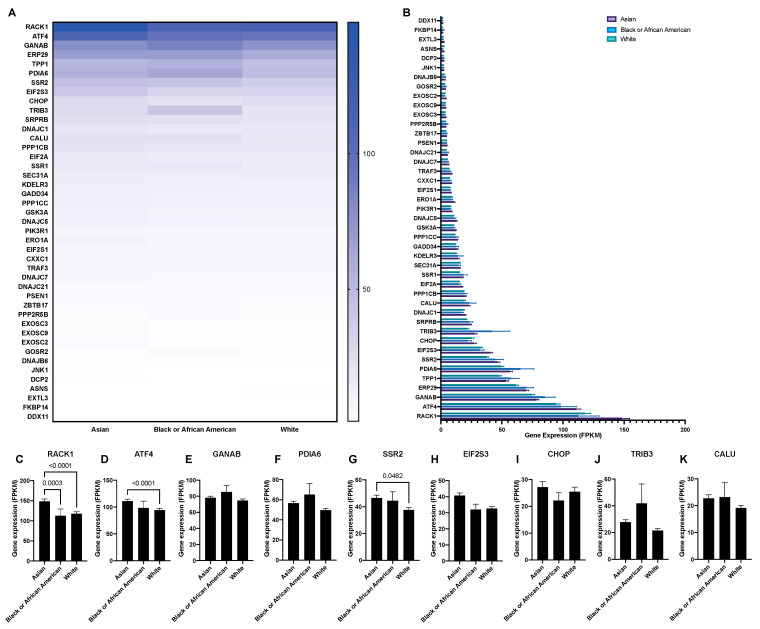
The mRNA expression of unfavorable UPR-associated markers per ethnic group shown in a heatmap (**A**) and a bar chart (**B**). The selected markers shown as individual bar charts (**C**–**K**).

**Figure 9 biology-10-00640-f009:**
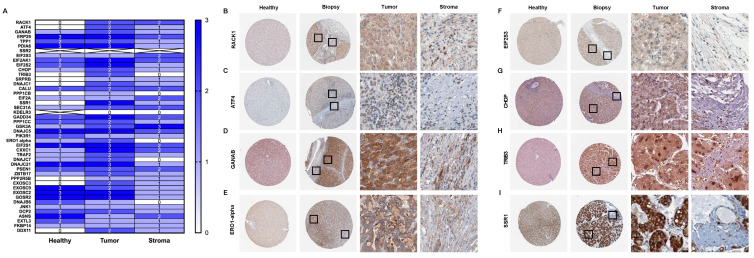
The histological scoring of healthy, stromal, and tumoral patient tissue stained against unfavorable UPR-associated markers (**A**). The selected markers are shown as individual images of healthy tissue and HCC biopsy (**B**–**I**).

## Data Availability

All data used in this study (transcriptomics data from 365 patients, healthy and HCC liver biopsies) were derived from The Cancer Genome Project (https://www.cancer.gov/tcga, accessed on 14 January 2021) and the Human Protein Atlas (https://www.proteinatlas.org, accessed on 14 January 2021) publicly available databases which are referred to in the manuscript according to the database guidelines.

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
