# Peer review of "Exploring the Role of Endoplasmic Reticulum Stress in Hepatocellular Carcinoma through mining of the Human Protein Atlas"

_biology, 2021, doi:10.3390/biology10070640_

Round 1
Reviewer 1 Report
Thanks for the answers to my comments.
Author Response
We thank the reviewer for their input on our manuscript.
Reviewer 2 Report
Based on the author's response, I remain unconvinced that this is a substantial advance to warrant publication in this high impact journal.
Author Response
We understand the reviewer's concerns, however we would like to point out that manuscript has been transferred from Cancers to Biology, a journal with a lower impact factor (3.7).
Reviewer 3 Report
Manuscript ID: Biology-1282419
Exploring the role of endoplasmic reticulum stress in hepatocellular carcinoma through mining of the Human Protein Atlas
The following points still need to be fixed.
- Line 153, SSR2 is not correct. Maybe CHOP. Figure 7I should be 7K.
- Lines 155-156, “the same markers as with age categories” may not be correct.
- Lines 233-237, since BiP is an ER lumenal protein, the description of this sentence should be modified. DnaJ proteins in the cytosol normally do not interact with BiP.
- Line 98, “strongest staining intensity” is still not clear. E.g., if the staining intensity of tissues treated with strong chemical ER stressors was scored as 3, it is reasonable. In Fig. 9A, the expression in tumor, healthy and stroma of some genes are all estimated as 1. How they are estimated as 1 (and e.g., not 2) is still unclear.
Author Response
Line 153, SSR2 is not correct. Maybe CHOP. Figure 7I should be 7K.
- We thank the reviewer for pointing this out, this has now been corrected.
Lines 155-156, “the same markers as with age categories” may not be correct.
- This has been corrected.
Lines 233-237, since BiP is an ER lumenal protein, the description of this sentence should be modified. DnaJ proteins in the cytosol normally do not interact with BiP.
- This has now been modified, and expanded with an example of DnajC1 interacting with BiP through binding its cytosolic domain to ribosome 80S.
Line 98, “strongest staining intensity” is still not clear. E.g., if the staining intensity of tissues treated with strong chemical ER stressors was scored as 3, it is reasonable. In Fig. 9A, the expression in tumor, healthy and stroma of some genes are all estimated as 1. How they are estimated as 1 (and e.g., not 2) is still unclear.
- As mentioned, when scoring the antibody staining intensity on the liver biopsies, undetectable staining corresponded with 0, and this was used as a reference. The scale of 1-3 was used to score the staining intensity of detectable staining. Thereby, each author independently scored 5 images stained with the same antibody, after which an average score was generated. This was now further clarified in the method section. While we agree that this is not a precise quantitative method, we acknowledge this in the manuscript and present this data as a semi-quantitative estimation of the UPR marker expression localization.
This manuscript is a resubmission of an earlier submission. The following is a list of the peer review reports and author responses from that submission.
Round 1
Reviewer 1 Report
I believe that the approach taken by the authors is interesting but not very untranslational. I think that beside the UPR associated proteins there are other stress response proteins (ERAD, ubiquitination, de-ubiquitination, etc..) that could be analysed taking advantage of the Human Atlas database. Anther important aspect which is not clear to me how the selected genes were validated, did the authors used the TMA technology or WB? This data is missing but I believe is important to include.
Reviewer 2 Report
Pavlović and colleagues performed an analysis of the human protein atlas database to identify changes in expression of ER stress markers in human liver cancer tissues with overall survival. They identified 44 ER stress genes linked to poor survival. Curiously these genes did not change much between stage I and stage IV disease but high expression of ER stress markers in general was linked with worse survival. It is not necessarily a new idea and the study is generally descriptive. Overall there are no concerns, but it is not clear whether this amount of work is sufficient to meet the high calibre of the journal Cancers.
Reviewer 3 Report
Manuscript ID: cancers-1240180
Exploring the role of endoplasmic reticulum stress in hepatocellular carcinoma through mining of the Human Protein Atlas
Endoplasmic reticulum (ER) stress and unfolded protein response (UPR) play important roles in the initiation and progression of hepatocellular carcinoma (HCC). To utilize as therapeutic targets or biomarkers of HCC, the authors evaluated the prognostic value of ER stress components using the Human Protein Atlas database. They identified 44 ER-stress-associated proteins as unfavorable prognostic markers in HCC.
The following points are required to be answered to clarify the main claim of the manuscript.
- In Figure 7 and in the result (line 148), the authors describe that the expression of SSR2 gene was different between ages. However, from the heatmap and the chart shown in Figs. 7A and B, the gene showing difference between ages seems to be CHOP, and not SSR2.
Furthermore, the ranges of error bars seem different in Fig. 7B and Figs. C-K.
- In the discussion, lines 222-235, the authors discuss on exosome, the extracellular vesicle. However, the genes encoded by EXOSC2, 3, and 9 are components of the exoribonuclease complex exosome, that processes and degrades RNA. This paragraph should be corrected.
- The methods how the UPR genes were selected from the online databases are not clear (lines 74-75). How were the exosome complex components selected (Fig.2)?
In addition, for example, although the gene HERP is known to be regulated by or interact with other genes, there are no lines between HERP and other genes (Fig. 2).
- The scheme shown in Fig. 1 is incorrect.
- The criteria how the intensity of staining was scored as 3 and others are not clear (lines 97-98).
- Most of the Hsp40/DnaJ proteins identified (lines 215-216) localize in the cytosol or the nucleus. Their functions and how they are involved in the HCC prognosis should be discussed.
- Explain briefly on the 1 gene identified as a favorable prognostic marker.
- A list should be supplemented depicting the gene symbols, gene or protein names, and their expected function or ontology.